# Effective Utilization of Copper Slag for the Production of Geopolymer Concrete with Different NaOH Molarity under Ambient Curing Conditions

Nagarajan Arunachelam [1], Jeyaprakash Maheswaran [2,*], Maheswaran Chellapandian [1] and Togay Ozbakkaloglu [3]

1  Department of Civil Engineering, Mepco Schlenk Engineering College, Sivakasi 626005, India
2  Department of Civil Engineering, St Xavier's Catholic College of Engineering, Nagercoil 629003, India
3  Department of Civil Engineering, Texas State University, San Marcos, TX 78666, USA
*  Correspondence: drjmaheswaran@gmail.com

**Abstract:** In spite of the considerable research on the mechanical and durability properties of geopolymer concrete, its widespread applicability is hindered due to the difficulties involved in achieving ambient curing conditions and awareness of the effective utilization of industrial by-products. This study investigates the physical and microstructure characterization of sustainable geopolymer concrete (GPC) developed with copper slag as a replacement for fine aggregate. In total, forty-four geopolymer concrete mixtures were prepared to examine their fresh and hardened properties. Four different NaOH molarities (10, 12, 14 and 16) and the replacement levels of copper slag, ranging from 0 to 100% with an increase of 10%, were considered as variables in this research. The study parameters examined includes the fresh (slump) and hardened concrete properties. Additionally, the microstructural characterization for different mixes were studied using the Fourier Transform Infrared Spectroscopy (FTIR), Electron Dispersive Spectrum (EDS) analyses and Scanning Electron Microscope (SEM). Results indicated that replacing fine aggregate with copper slag up to 100% showed no strength reduction. Increasing the molarity of the NaOH solution to 16M led to an increased strength of about 35% compared to the concrete with 10 M in all the mixes. The microstructural analysis performed using SEM/EDS and FTIR showed that a cohesive and fully compact geopolymer matrix was achieved together with the use of low-calcium fly ash and copper slag under ambient curing conditions.

**Keywords:** alkaline activators; copper slag; geopolymer concrete; low-calcium fly ash; microstructure analysis; NaOH molarity

## 1. Introduction

Concrete is frequently utilized as a construction material for the development of infrastructure due to its superior mechanical qualities and low prices. However, using cement as the primary binding material causes a massive carbon footprint and leads to several environmental issues. It is estimated that more than 10 million tons of concrete are produced globally per annum [1]. In 2016, 8% of the earth's carbon dioxide emissions were due to the production of cement [2]. Moreover, the manufacture of Portland cement requires a large number of raw ingredients (e.g., the production of two billion tons requires about three billion tons of raw materials) [3]. Therefore, to decrease the negative effects of $CO_2$ emissions, it is essential to develop an effective alternative construction material which could potentially replace cement in concrete. An alternative to cement-based concrete, geopolymer concrete (GPC), uses additional cementitious elements such as ground granulated blast furnace slag (GGBS), fly ash, other recycled or industrial waste materials and alkaline activators (NaOH/KOH and $Na_2SiO_3$) [4–6].

Geopolymerization is a complex process and involves three major stages including (a) dissolution of Si and Al from the source material in the alkaline environment, (b) reorganization and diffusion of dissolved ions with the formation of small coagulated structures and (c) polycondensation of the coagulated structure resulting in the formation of hydration products [7,8]. Fernández-Jiménez and Palomo [9] studied the reaction mechanism and the formation of N-A-S-H gel during the reaction between fly ash and an alkaline activator solution. The following are the influence of Si, Al and Na in the formation of N-A-S-H gel:

- Si elements present in the source material and silicate solution initially participate in the gel formation based on the zeolitic nuclei through dissolution during the early polymerization, which results in silica-rich gel development.
- Al present in the source material is released into the solution, which improves the reactivity. The dissolved Al enters into the silica gel structure to impart better stability to the polymerization.
- Na from the activator solution acts as a balancer by altering the Al present in the gel to enhance the stability, thereby creating a dense matrix to bind the oxygen and water molecules.

To facilitate the polymerization process, a quick interaction of silica and alumina is essential under alkaline conditions, and this phenomenon establishes a Si-O-Al-O bond called a 3D (three-dimensional) polymeric bond chain [10]. It is worth mentioning that the production of GPC requires only 3/5th of the energy and contributes about 85% less $CO_2$ emissions when compared to the production of Portland cement [11]. Moreover, GPC promises superior durability performance compared to cement-based concrete because GPC achieves a better particle packing [5].

In addition to its mechanical and durability performance, GPC possesses good thermal resistance and could sustain high temperatures in the range of 1200 °C without any sudden degradation [12]. In the last decade, many researchers have shown interest in investigating the mechanical and durability characteristics of GPC derived from industrial by-products [4,13–17]. Nagajothi and Elavenil [18] studied the use of GGBS and fly ash to produce GPC. Their results indicated that 42 MPa compressive strength was observed within 40 h of curing using ambient conditions. Hardjito and Rangan [19] investigated the engineering properties of geopolymer concrete with high-volume fly ash. They reported that the behavior of the geopolymer concrete in terms of failure mode was similar to conventional concrete. Rangan [15] investigated the durability and mechanical characterization of geopolymer concrete with different test variables such as alkaline activator concentration and curing method, including rest period, temperature and duration. In addition to the studies mentioned above, several recent studies also highlighted the improved strength and durability properties of GPC [20–22].

In concrete, sand has been predominantly used as the fine aggregate and is mined from sources such as river beds, open pits, beaches, etc. At present, the global requirement of sand for the infrastructure industry is 3.7 billion tonnes [23]. However, with the start of significant infrastructure projects and ongoing improvements in the construction industry, this figure might skyrocket. Also, the mining process may cause serious environmental problems, such as (a) reduced groundwater recharge into aquifers, (b) soil degradation, (c) public annoyance due to sand excavation and transportation and (d) destruction of fauna and flora in the nearby regions. Hence, there should be an essential awareness to use an alternative material in concrete to preserve river sand [23–25]. Moreover, this ventures the use of industrial waste materials such as copper slag, pond ash, manufactured sand, and recycled sand as a substitute for river sand in concrete. Several previous studies have investigated the behavior of concrete containing copper slag and reported that the optimum level of replacement in cement concrete was 40% to 50% in place of natural river sand [26–34].

Brindha et al. [35] explored the durability of copper slag-based concrete as a replacement for sand up to 50% and determined the optimum level of replacement was 40% copper slag in place of natural river sand. Al-Jabri et al. [36] studied the properties of

copper slag concrete in which copper slag was used as a partial replacement for sand. They found that the optimum replacement level of copper slag was 50% for achieving similar performance to the control concrete. Palani et al. [37] replaced conventional river sand with copper slag up to 100% in steps of 10%. They concluded that the replacement of river sand with copper slag is effective even at higher levels. Sharma and Rizwan [38] examined the performance of a 20% copper slag substitute in self-compacting concrete (SCC). They concluded that the compression and splitting tensile strength had improved significantly. Furthermore, images from an SEM analysis revealed the formation of closely crowded and evenly dispersed white deposits (i.e., C-S-H gel) when conventional river sand is partially replaced with copper slag. According to another study, the mechanical and microstructural features of geopolymer concrete containing a high volume of copper slag are yet to be explored. The emphasis of our work is to study the effect of different NaOH molarities and high-volume copper slag addition on the mechanical and microstructural features of ambient-cured GPC.

Despite the considerable research available on geopolymer concrete, a very limited number of past studies have focused on ambient-cured GPC. There are no investigations that directly mention the use of high-volume copper slag instead of sand in the development of ambient-cured geopolymer concrete. This knowledge gap is due to the challenges involved in achieving the process of ambient curing conditions, which comprises adding more than one alkali activator or increasing the concentration of alkali activators for reaching the desired strength. The major objective of the present study is just to explore the workability, mechanical and microstructural behavior of high-volume copper slag-based ambient-cured GPC made with different molarities of a NaOH solution. The distinctive contributions made by this work are listed below:

(a) Developing a high-volume copper slag-based geopolymer concrete mix to replace river sand.
(b) Estimating the influence of different molarity levels of NaOH (10 M, 12 M, 14 M and 16 M) on the workability, strength and microstructural characteristics of ambient-cured geopolymer concrete.
(c) Understanding the strength and microstructural characteristics of sustainable geopolymer concrete with copper slag through the compression test, split tensile test, modulus of rupture, SEM with EDS analysis and FTIR analysis.

## 2. Experimental Investigation

The main variable considered in this study was the percentage of copper slag replacement for sand in geopolymer concrete. Also, the cast specimens were analytically verified through the prediction equation provided in the standards, and it was compared with the experimental results.

### 2.1. Materials
#### 2.1.1. Fly Ash

Class-F fly ash acquired from the TTPS, Tuticorin, India, was used as the major binding material in the production of sustainable GPC. The physical characteristics were analyzed using an SEM, and the visualization of particles was done using a magnification of up to 150 kX times. Figure 1a shows the SEM image of a fly ash sample, and the results reveal the existence of glassy spherical particles, which could help in enhancing the mixing and flowability of the developed concrete mix. Figure 1b depicts the sharp crystalline peaks of fly ash obtained from the X-ray Diffractometer (XRD) analyses. The presence of sharp peaks confirm the crystalline nature (i.e., mullite and quartz) of fly ash used in this study. In addition, a few amorphous stages can be found from the presence of a hump between the diffraction angle of 14° and 30° [39–41]. Table 1 lists the elemental oxide composition of the fly ash analyzed using X-ray Fluorescence (XRF). The results reveal a high content of alumina (14.8%), silica (69.6%) and iron (3.3%), which conforms to the ASTM guidelines [42].

In addition, the particle size distribution was analyzed for fly ash, and the average size of particles present in the fly ash sample was 11.93 μm with a standard deviation of 0.7 μm.

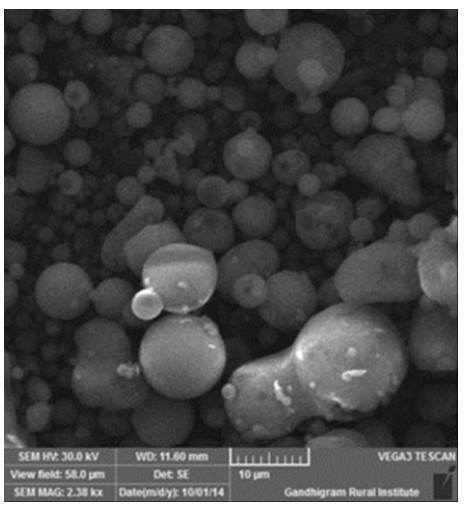

(**a**) SEM analysis

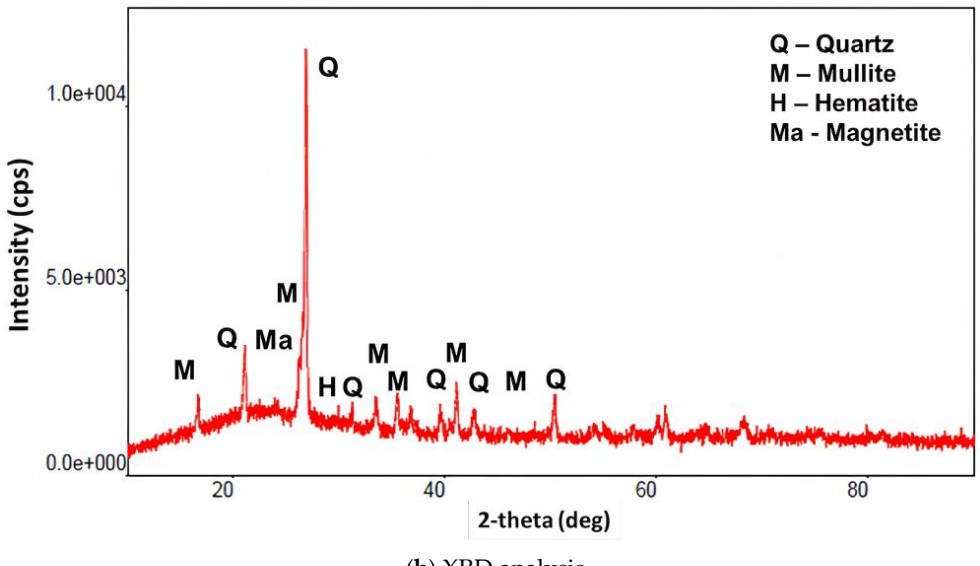

(**b**) XRD analysis

**Figure 1.** Microstructure characterization of fly ash.

**Table 1.** XRF analysis for binder and aggregate.

| Details of Oxide | Fly Ash | ASTM C618 Requirements % | Cu. Slag | River Sand |
|---|---|---|---|---|
| Silica ($SiO_2$) | 69.6 | | 44.4 | 95.5 |
| Alumina ($Al_2O_3$) | 14.8 | Minimum—70% | 3.20 | 0.90 |
| Iron oxide ($Fe_2O_3$) | 3.30 | | 42.9 | 0.80 |
| Calcium oxide (CaO) | 4.50 | | 2.40 | 0.26 |
| Sodium oxide ($Na_2O$) | 4.70 | | 0.03 | 0.00 |
| Magnesium oxide (MgO) | 1.60 | | 4.20 | 2.30 |
| Sulphur trioxide ($SO_3$) | 0.50 | Maximum—3% | 2.60 | 0.72 |
| Loss of Ignition | 0.90 | | 0.30 | 0.25 |

### 2.1.2. Copper Slag

A granular, glassy black element formed during the extraction of copper with the values of specific gravity, bulk density and water absorption measuring 3.58, 2045 kg/m$^3$ and 0.23%, respectively. It is important to note that the copper slag has an extremely high value of specific gravity owing to the presence of iron. Figure 2 depicts the particle size distribution graph for the copper slag, which resembles that of the river sand, and the results comply with the Zone II category. The SEM results for copper slag show that the particle shape is irregular (i.e., angular) with the presence of sharp edges (Figure 3a). The elemental oxide composition of copper slag was obtained from the XRF analysis, and the major constituents present such as $SiO_2$ and $Fe_2O_3$ are shown in Table 1. Moreover, the copper slag contains more than 80% silica and iron oxide. Similarly, from the XRD analysis, no major peaks were found due to its amorphous phase (Figure 3b).

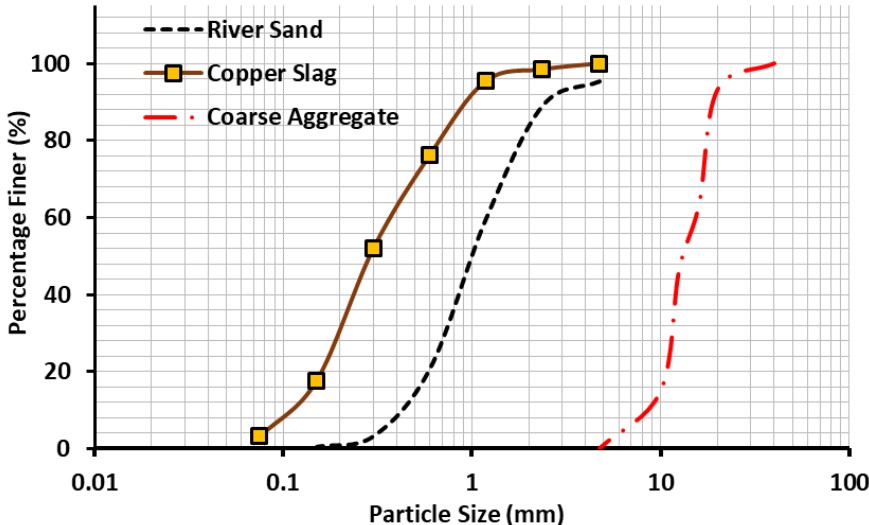

**Figure 2.** Particle size distribution comparison for aggregates used.

### 2.1.3. River Sand

GPC was made with locally available river sand that complies with the Zone II category. The particle size distribution graph of river sand is presented in Figure 2. The pattern of the particle distribution follows a smooth curve showing the well-graded nature of the fine aggregate. The fineness modulus for the river sand was assessed to be 2.57, whereas the water absorption and specific gravity values were 0.94% and 2.52, respectively. From the XRF analysis, $SiO_2$ was found to be the major constituent occupying more than 95% compared to the other oxide components.

### 2.1.4. Coarse Aggregate

In this work, blended coarse aggregates (i.e., size = 10 mm and 20 mm) were used in the manufacture of GPC. From the physical characterization tests, it was determined that the specific gravity value, bulk density, void ratio and water absorption values were 2.74, 1420 kg/m$^3$ and 0.45%, respectively. Figure 2 depicts the particle distribution graph, which confirms that the well-graded nature of aggregates follows an "S-curve" distribution form.

### 2.1.5. Alkaline Activator Solution

NaOH solids in the form of flakes with 98% purity and a specific gravity value of 2.13 were used. The NaOH solution was prepared 24 h before the time of casting and was introduced or mixed with $Na_2SiO_3$ at the time the concrete was prepared. In the case of the sodium silicate solution, the chemical configuration observed is as follows: $Na_2O$: 14.8%, $SiO_2$: 29.6% and $H_2O$: 55.6%.

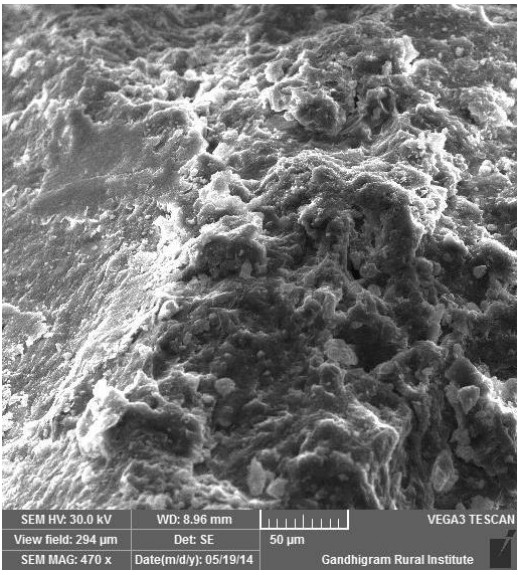

(**a**) SEM Analysis

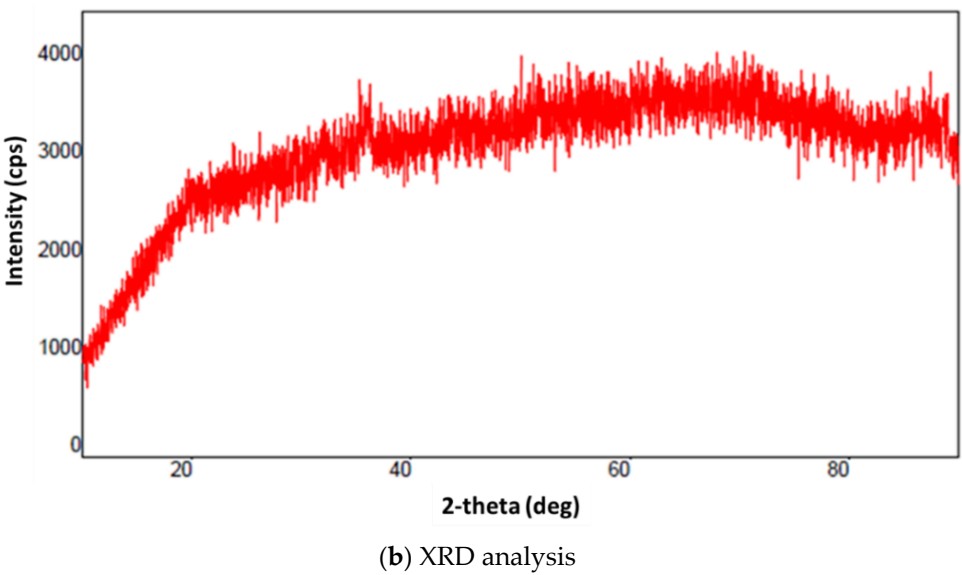

(**b**) XRD analysis

**Figure 3.** Microstructure characterization of copper slag.

Two-component alkaline activators containing sodium hydroxide (NaOH) and sodium silicate ($Na_2SiO_3$) liquids in the ratio of 1:2.5 were deployed for the casting of copper slag GPC specimens [14,34–36,43–48]. Alkaline activators aid gel formation by allowing silica and alumina to dissolve and generate hydration products (i.e., Na-Al-Si-H gel). To obtain the required molarity of sodium hydroxide, flakes were acquired locally and dissolved in mineral water [34]. The quantity of NaOH solids and water required to produce the desired molar solution of NaOH is inferred from previous studies [16]. For example: for the preparation of 1 L of 10 molarity NaOH solution, 306 g of NaOH salts were dissolved in 694 g of distilled water. Similarly, a sodium hydroxide solution with four different molarity levels (10 M, 12 M, 14 M and 16 M) was prepared and investigated as a part of this work. Another alkaline solution which is sodium silicate ($Na_2SiO_3$) was obtained in liquid form and used for mixing. Regarding the use of distilled water for the GPC mix, most of the previous studies adopted the molarity-based concept. Hence, only distilled water, which is free from impurities/salt concentration, was used for the mix development. It is worth mentioning that the tap water available in certain localities contains high salinity, which can affect the molarity of the design mix.

### 2.1.6. Mix Proportion for GPC

There are no standards available that describe the mixing procedure for geopolymer concrete. Therefore, the mix design for the geopolymer control concrete was prepared using a trial and error method. The procedure for the preparation of different geopolymer concrete mixes with copper slag and NaOH molarity levels is schematically represented in Figure 4. Different trails of the control GPC were performed to attain a mean compressive strength of 40 MPa. Table 2 highlights the details of the GPC mixes studied with varying levels of copper slag replacement. In total, 44 GPC mixes were evaluated with different levels of copper slag and NaOH molarity. For all the mixes, the ratio of alkaline liquid to fly ash ratio was maintained constant at 0.4. To accomplish the alkaline activator preparation, the required molarity of the NaOH solution was mixed with the $Na_2SiO_3$ solution. During the mixing of concrete, the dry materials such as aggregates and fly ash were initially mixed in the mixture pan, and then the activator solution was added. To improve the workability, 4% sulpho-napthalene formaldehyde (SNF)-based water reducer was used for the manufacturing of GPC. In addition, it is worth mentioning that the dosage was finalized from a few previous studies which have provided super-plasticizers up to 6% in GPC [49,50].

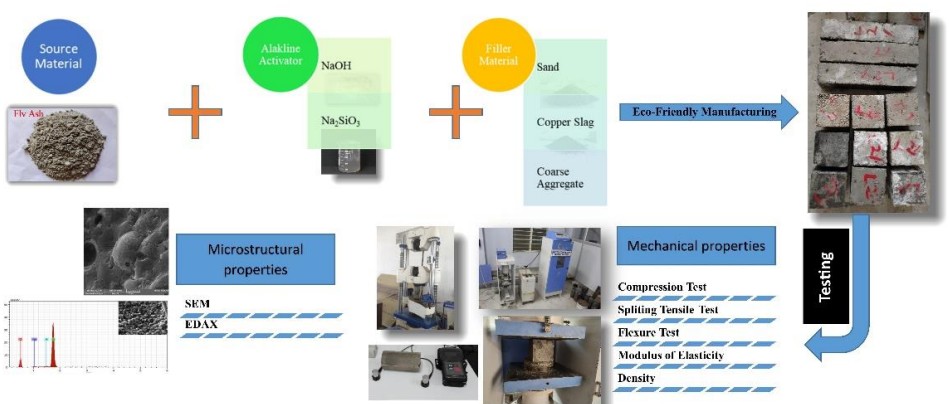

**Figure 4.** Procedure for the production of high-volume copper slag-based GPC.

**Table 2.** Test matrix and details of the mix design.

| Specimen ID | Mix Proportions (kg/m$^3$) | | | | | | | Remarks |
|---|---|---|---|---|---|---|---|---|
| | **Fly Ash** | **RS** | **CS** | **CA** | **NaOH** | **Na$_2$SiO$_3$** | **SP** | |
| GPC_CC | 460 | 613 | 0 | 1139 | 53 | 158 | 19.2 | Control mix with the molarity of sodium hydroxide (NaOH) varied as 10 M, 12 M, 14 M and 16 M |
| GPC_CS10 | 460 | 551 | 94 | 1139 | 53 | 158 | 19.2 | |
| GPC_CS20 | 460 | 490 | 188 | 1139 | 53 | 158 | 19.2 | |
| GPC_CS30 | 460 | 429 | 282 | 1139 | 53 | 158 | 19.2 | |
| GPC_CS40 | 460 | 367 | 376 | 1139 | 53 | 158 | 19.2 | Fine aggregate replaced with copper slag in levels from 0–100%. In addition, the molarity of sodium hydroxide (NaOH) is varied as 10 M, 12 M, 14 M and 16 M |
| GPC_CS50 | 460 | 306 | 470 | 1139 | 53 | 158 | 19.2 | |
| GPC_CS60 | 460 | 245 | 564 | 1139 | 53 | 158 | 19.2 | |
| GPC_CS70 | 460 | 183 | 658 | 1139 | 53 | 158 | 19.2 | |
| GPC_CS80 | 460 | 122 | 752 | 1139 | 53 | 158 | 19.2 | |
| GPC_CS90 | 460 | 61 | 846 | 1139 | 53 | 158 | 19.2 | |
| GPC_CS100 | 460 | 0 | 940 | 1139 | 53 | 158 | 19.2 | |

### 2.1.7. Ambient Curing Mechanism for GPC

The cast specimens were de-molded and maintained at room temperature to facilitate the process of ambient curing. The mechanism of ambient curing is utilized to increase the feasibility of geopolymer concrete for further practical implementations in the construction of infrastructural elements. The mechanism of polymerization by ambient curing takes place in three steps. The first step is the dissolution of Si and Al from the source material in the alkaline environment. The second step is the reorganization and diffusion of dissolved ions with the formation of small coagulated structures. In the final step, polycondensation takes place in the coagulated structure and results in the hydrated products, i.e., the formation of N-A-S-H gel during the reaction between the fly ash and alkaline activator solution [7,8].

## 3. Fresh and Hardened Properties of GPC

Geopolymer concrete with copper slag at different sand replacement levels was studied for its mechanical performance using cube, cylindrical and prism specimens of sizes 100 mm cube, 150 mm dia $\times$ 300 mm height and 100 mm $\times$ 100 mm $\times$ 500 mm, respectively. The percentage of copper slag replacement in place of sand and the molarity of the NaOH solution in the developed GPC were the main variables considered in this study.

### 3.1. Workability of Concrete

The workability of the freshly produced GPC with different NaOH molarities was evaluated using the slump cone test. The first mix (GPC CC), which is used as the benchmark for comparison and has 0% replacement of river sand, attained a 75 mm slump value. The workability achieved for this benchmark mix can be categorized under the medium slump range. When a higher dosage of copper slag is introduced into the GPC mix, the value of workability reduces to a significant extent. It is worth noting that the concrete mix became considerably firmer due to the presence of copper slag in place of sand, which made handling and filling the GPC mix into the molds much more difficult. This is attributed due to the shape of copper slag particles (i.e., irregular nature) present in the developed mix. Due to this, the workability of GPC was found to show a considerable reduction when the volume of copper slag was at higher levels (Figure 5). The decrease in workability may also be attributed to the high density of the raw materials used in the mix, i.e., the specific gravity of copper slag. As a result, the density of concrete showed an incremental increase of 9.3% from the value of 2408 kg/m$^3$ to 2632 kg/m$^3$ when copper slag was replaced with 100 percent river sand. The changes in NaOH molarity of the GPC mix from 10 M to 16 M had a significant effect on the workability of the developed mix. When the molarity of the NaOH solution increases, the slump value reduces due to the increased evolution of latent heat due to the polymerization reaction. This increase in sodium hydroxide (NaOH) molarity could result in a faster setting process of geopolymer concrete. Furthermore, this could have an impact on the ambient curing nature of geopolymer concrete.

### 3.2. Compressive Strength

Figure 6 depicts the results for standard cube specimens of size 150 mm tested at 7 and 28 days of ambient curing. From the test results, it can be inferred that the compressive strength of ambient-cured geopolymer concrete was influenced by three factors namely the (a) molarity of NaOH, (b) curing time and (d) percentage of copper slag used. In addition to the above-mentioned parameters, the particle size and surface area of the fly ash also influence the compressive strength of the GPC [16]. For the fly ash-based geopolymer concrete with NaOH and Na$_2$SiO$_3$ as the alkaline solutions, the main by-product is the development of a hydration compound (i.e., Na-Al-Si-H gel) [51–53]. This Na-Al-Si-H gel is responsible for the development of strength, which progresses over time under ambient curing conditions. Three major elements including Si, Al, and Na play a major role in the development of strength and come either from the fly ash or from the activator solution [54].

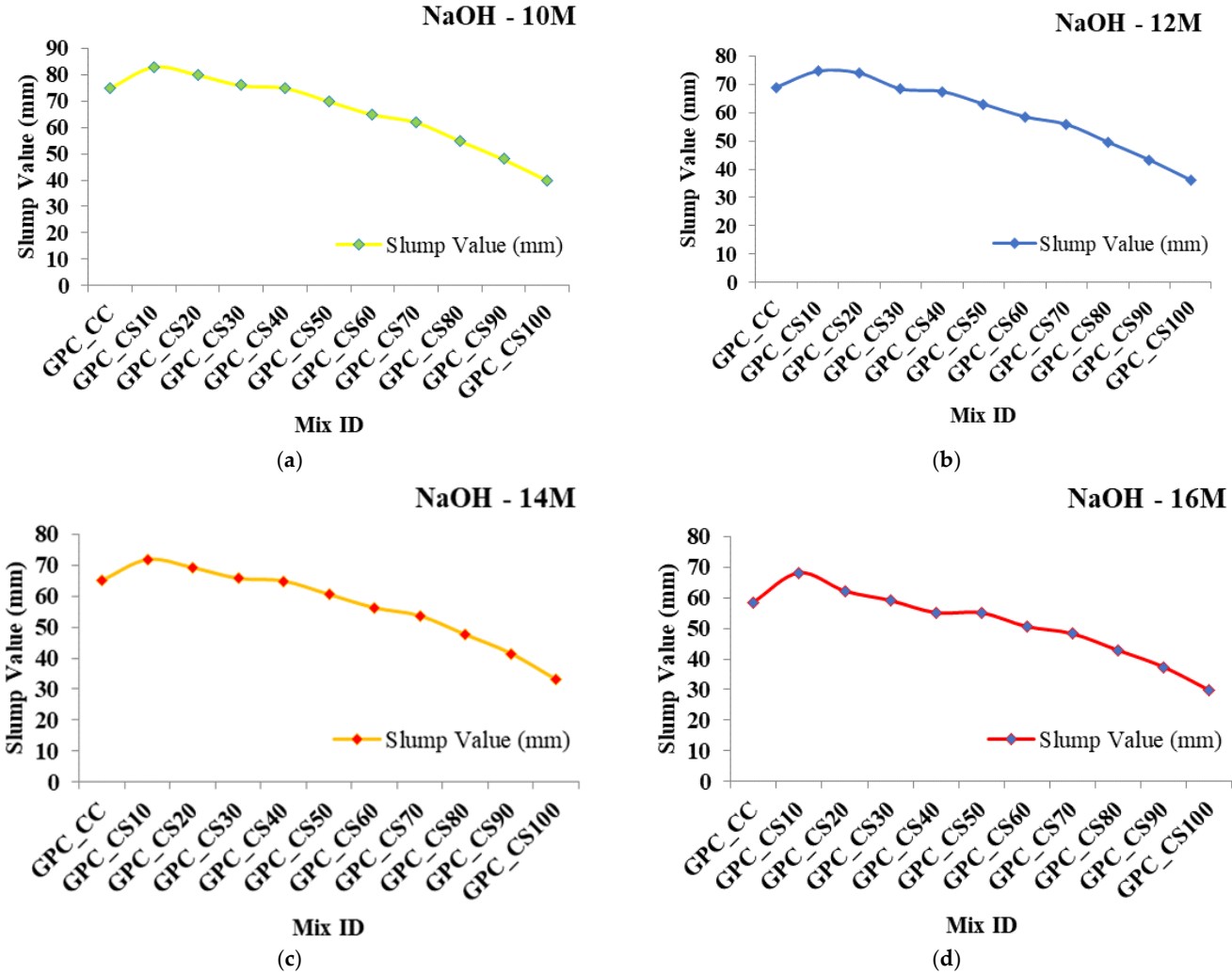

**Figure 5.** Slump values of GPC mixed with copper slag (**a**) 10 M, (**b**) 12 M, (**c**) 14 M, (**d**) 16 M.

When compared to the compressive strength attained after 28 days of ambient curing, the strength attained for all GPC mixes after 7 days of ambient curing was approximately 30%. This low strength at initial phases is due to the low heat of hydration achieved, whereas the polymerization process requires a high latent heat [10]. However, the scenario improves with an increase in curing time. Similarly, at 28 days of ambient curing, most of the mixes had a higher compressive strength required for practical applications. Specifically, the use of copper slag at high levels of river sand replacement results in creating sustainable construction practices and could also aid in enhancing the polymerization reaction due to their pozzolanic property [14].

The compressive strength of GPC with 100% copper slag replacement was significantly higher than that of the control GPC specimens. For the 10 M GPC mix, the replacement with 100% copper slag helped to enhance the 28-day compressive strength by 46.9% when compared to the control specimen. Similarly, the mix with a NaOH molarity of 12 M, 14 M and 16 M had an enhanced strength percentage of 51.8%, 53.6% and 51.3%, respectively. The cementitious properties of copper slag, which react with the NaOH solution to generate the alumino-silicate gel, are the prime reason for the rise in the compressive strength of the GPC [55,56]. Among the 44 mixes studied, the mix GPC_CS100, which contains 100% copper slag and 16 M of NaOH solution, showed the highest 28-day compressive strength value of 46.0 MPa, eliminating the need for heat curing of GPC. This mix attained a strength value of 13.6 MPa at 7 days in the curing period, which was about three times lower than the 28-day strength. Therefore, the ambient curing of GPC specimens results in the

development of strength through alumino-silicate gel over time similar to the OPC-based concrete [17,57–60].

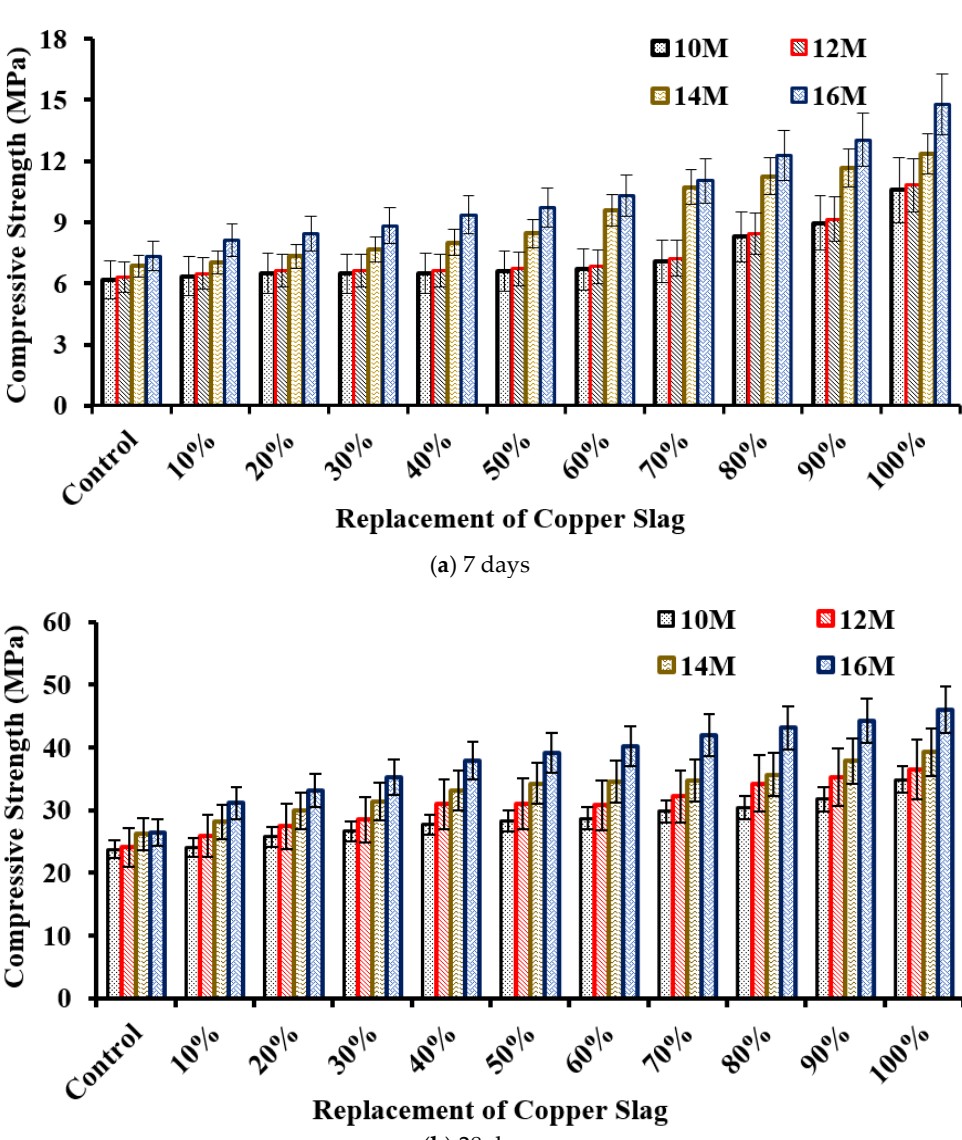

(**a**) 7 days

(**b**) 28 days

**Figure 6.** Comparison of compressive strength results for different molarity levels of NaOH.

### 3.3. Split Tensile Strength

The split tensile strength test was performed on cylindrical specimens of size 150 mm diameter and 300 mm height and tested on the 7th and 28th day of the ambient curing conditions. The split tensile test of GPC cylinders was performed following the Indian Standards. The results obtained from the tests are compared with the standard design equations proposed by different countries. Table 3 shows the design equations specified by different standards such as the ACI 318-11 [61], EC-04 [62], JCI-08 [63], JSCE-07 [64] and NZS-06 [65] for the prediction of the split tensile strength. Experimental split tensile results for the GPC with different molarities were compared with the theoretical predictions and presented in Figure 7. From the figure, it can be concluded that all the design equations underestimated the test values. The prediction from the ACI 318-11 [61] showed a close correlation when compared to the other design equations. From the split tensile test results shown in Figure 7, it can be inferred that the use of high-volume copper slag in geopolymer concrete increases its tensile strength. Also, this trend of increase in tensile strength with

the increased replacement levels of copper slag was found to be similar when compared to the compressive strength of the GPC mix.

Among the 44 mixes studied, the mix GPCS100, which contains 100% copper slag and 16 M of NaOH solution, showed the highest 28-day split tensile strength value of 2.72 MPa. This splitting tensile strength value was found to be 1.3 times higher when compared to the control specimen. The pozzolanic activity of copper slag in the GPC was primarily responsible for this high strength [14]. In addition to the copper slag replacement, the molarity of NaOH also had a significant impact on enhancing the tensile strength of the prepared geopolymer concrete mixes [49]. For the control mix with 10 M NaOH solution, an average split tensile strength for cylindrical GPC specimens was determined to be 1.63 MPa. By increasing the molarity to 12 M, 14 M and 16 M, the split tensile strength value increased by 8%, 19% and 24.6%, respectively, when compared to a control specimen with 10 M NaOH solution (CC10). It is worth mentioning that the geopolymer concrete mixes tested had a high level of polymeric bonding between the geopolymer paste and the filler materials in the GPC. According to the results, no aggregates were pulled out during the split tensile test, which is likely to happen when conventional concrete is made with OPC because of the chemical bond between the aggregates and the alkaline liquid [66].

**Table 3.** Design equations for predicting the tensile strength and elastic modulus.

| Guidelines Name | Elastic Modulus | Flexural Strength | Split Tensile Strength |
|---|---|---|---|
| ACI 318-11 [61] | $E_c = 4.73 \sqrt{f'_c}$ | $f_r = 0.62 \sqrt{f'_c}$ | $f_{st} = 0.53 \sqrt{f'_c}$ |
| EC-04 [62] | $E_c = 22(f_{cm}/10)^{0.3}$ | $f_r = 0.435 f'_c{}^{2/3}$ | $f_{st} = 0.30 (f'_c)^{2/3}$ |
| JSCE-07 [64] | $E_c = 4.7 \sqrt{f'_c}$ | - | $f_{st} = 0.44 \sqrt{f'_c}$ |
| JCI-08 [63] | $E_c = 6.3 f'_c{}^{0.45}$ | - | $f_{st} = 0.130 (f'_c)^{0.85}$ |
| NZS 3101 [65] | $E_c = 3.32(\sqrt{f'_c}) + 6.9$ | $f_r = 0.60 \sqrt{f'_c}$ | $f_{st} = 0.44 \sqrt{f'_c}$ |
| IS456-2000 [67] | $E_C = 5000 \sqrt{f_{ck}}$ | $f_r = 0.7 \sqrt{f_{ck}}$ | - |
| CSA A23.3-04 [68] | $E_c = 4.5 \sqrt{f'_c}$ | $f_r = 0.60 \sqrt{f'_c}$ | - |

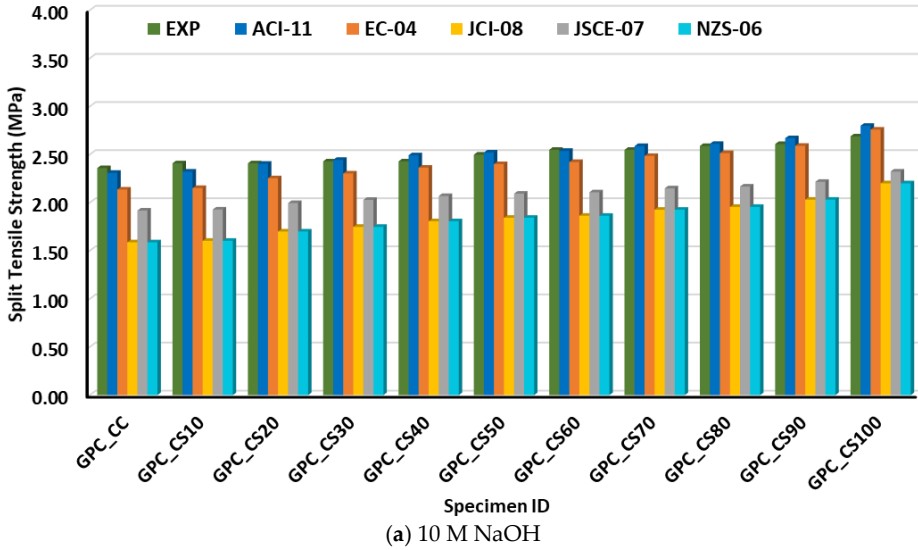

(**a**) 10 M NaOH

**Figure 7.** *Cont.*

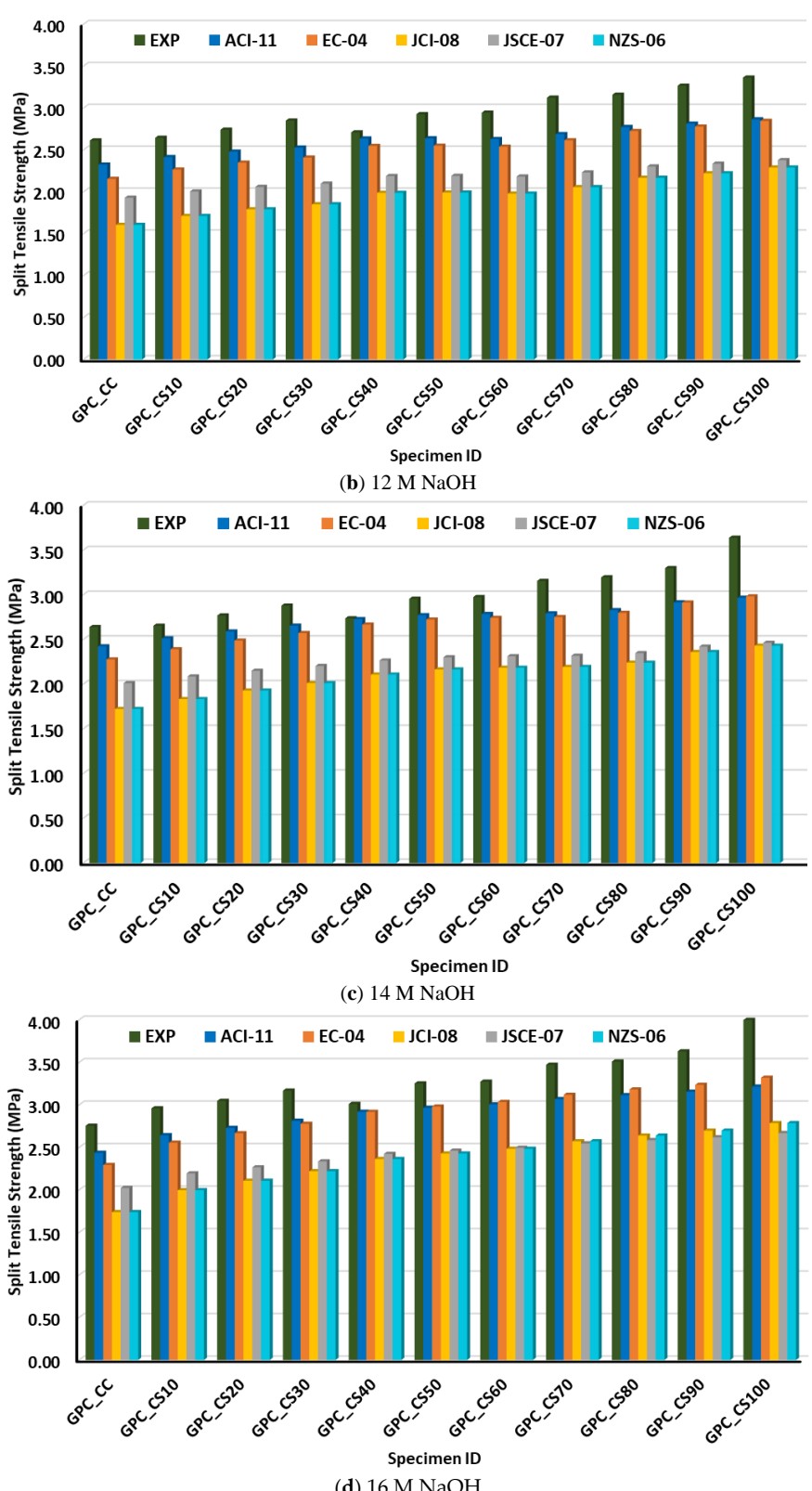

**Figure 7.** Comparison of split tensile strength results for different molarity levels of NaOH.

### 3.4. Flexural Strength

The flexural strength test for the developed GPC with different NaOH molarities and copper slag was performed on the 28th-day ambient-cured GPC of size $100 \times 100 \times 500$ mm. Figure 8 depicts a comparison of the experimental and theoretical flexural strength values

of the GPC specimens. The theoretical value of the flexural strength predicted using the design equation in IS 456-2000 [67] showed a close correlation with the test results. It can also be inferred that the addition of copper slag as a substitute for sand and the molarity of the NaOH solution have a strong influence on the flexural strength of the geopolymer concrete specimen. The flexural strength of the control GPC specimens with 10 M NaOH was found to be 2.56 N/mm$^2$. With the increase in molarity to 16M, the flexural strength increased by 13.2% when compared to the CC10. The flexural strength was found to exhibit an increasing trend with an increase in the substitution of river sand for copper slag, similar to the findings of compressive strength and split tensile strength. The specimens with copper slag as a fine aggregate showed a 24.6% increase in flexural strength compared to the control specimens.

### 3.5. Elastic Modulus

The elastic modulus (E) of the GPC specimens was measured by testing the cylindrical specimens of 150 mm diameter and 300 mm height that were cast for two specimen series including the (i) controls (GPC_CC) and (ii) GPC with 100% copper slag. In these two mixes, the molarity of NaOH varied from 10 M to 16 M. The cast specimens were tested with a compressometer to measure the axial and lateral strain. The experimental result was compared with the theoretical modulus of elasticity values predicted using the design equations in Table 3. Comparing the experimental and theoretical values shown in Figure 9, it can be concluded that the equation used for calculating the modulus of elasticity of the GPC specimens was overestimated. The experimental values of the elastic modulus for GPC are closely captured using the equation suggested by the CSA-04 [68].

### 3.6. Ultrasonic Pulse Velocity

Ultrasonic pulse velocity (UPV) is an indirect way of assessing the quality of GPC using major observations such as cracking, the homogeneous nature of the evaluated material and the existence of inner faults. The non-destructive test is carried out on a 100 mm cube specimen following the IS 13311-1992(1) after the curing period. The test results show that the velocity range was between 3.65 km/s and 3.90 km/s. It is worth mentioning that the presence of the gel water produced during the polymerization reaction was observed to cause the ambient curing specimen to have decreased ultrasonic pulse velocity values [69–72]. The obtained results indicate that the pulse velocity increases as the amount of copper slag in the concrete increases (Figure 10).

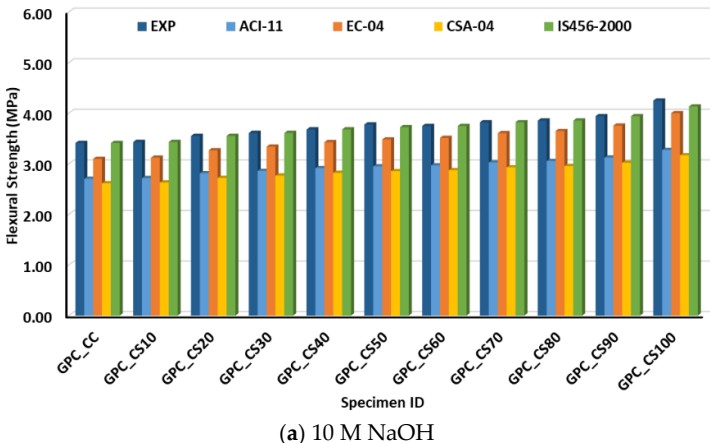

(**a**) 10 M NaOH

**Figure 8.** *Cont.*

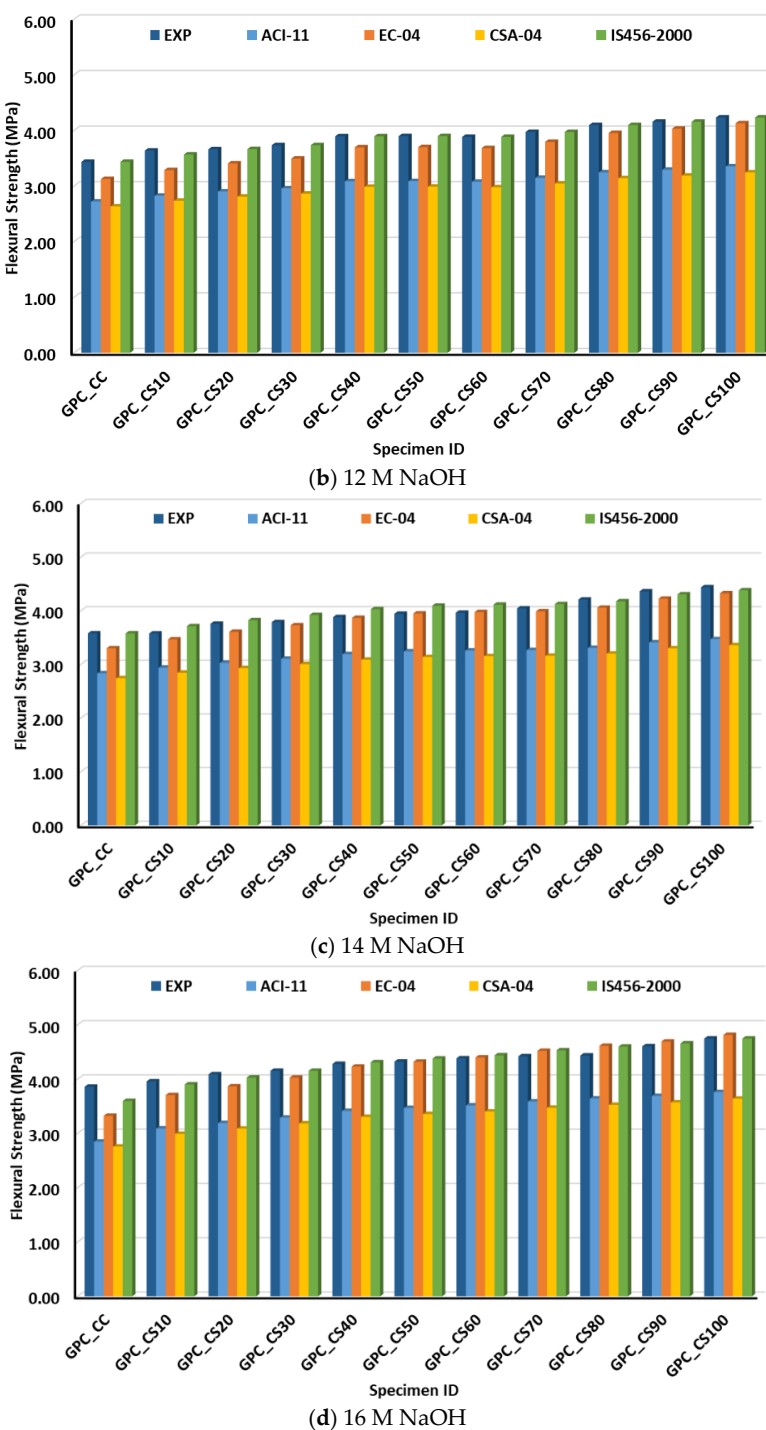

**Figure 8.** Comparison of flexural strength results for different molarity levels of NaOH.

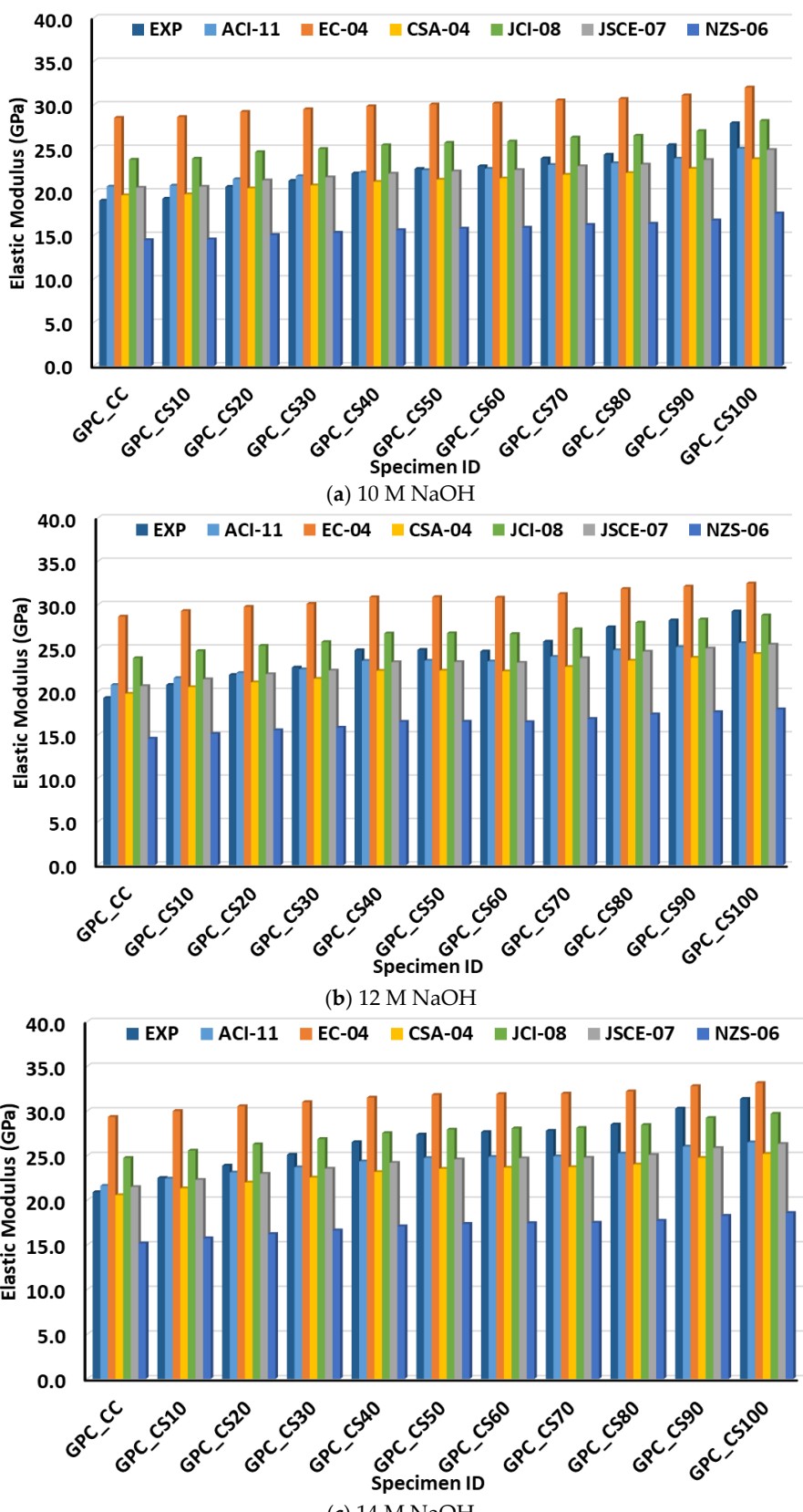

(**a**) 10 M NaOH

(**b**) 12 M NaOH

(**c**) 14 M NaOH

**Figure 9.** *Cont.*

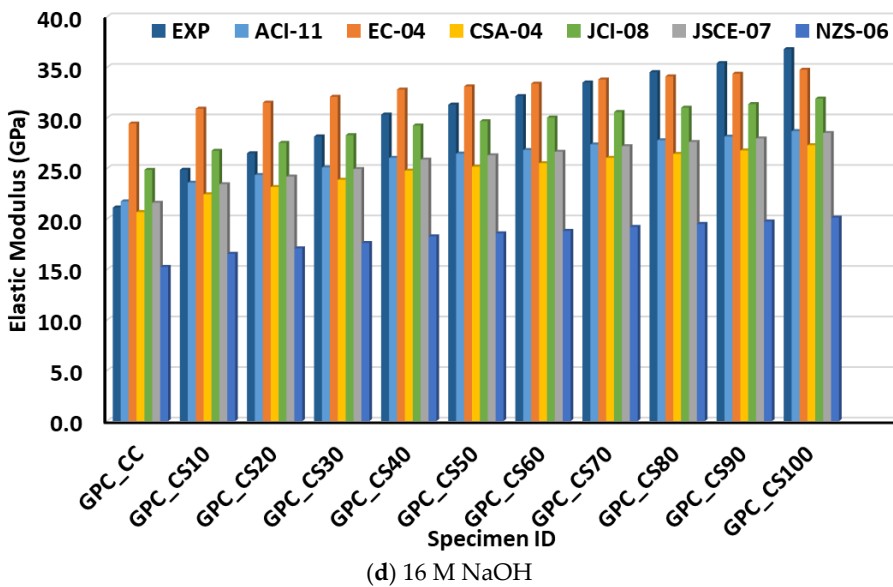

(**d**) 16 M NaOH

**Figure 9.** Elastic modulus of GPC mixed with different NaOH molarities.

The UPV value measured for the 10 M NaOH-based GPC specimen (GPC_CS100) was 3.65 km/s. Similarly, for the geopolymer concrete prepared with the 12 M NaOH solution, the pulse velocity was determined to be 3.70 km/s. In ambient curing conditions, the measured pulse velocity for the 14 M NaOH solution was 3.65 km/s, which was found to be similar to the 10 M NaOH-based GPC specimen (GPC_CS100). The UPV value of 3.89 km/s was observed on the ambient-cured geopolymer concrete specimens prepared using the 16 M NaOH solution.

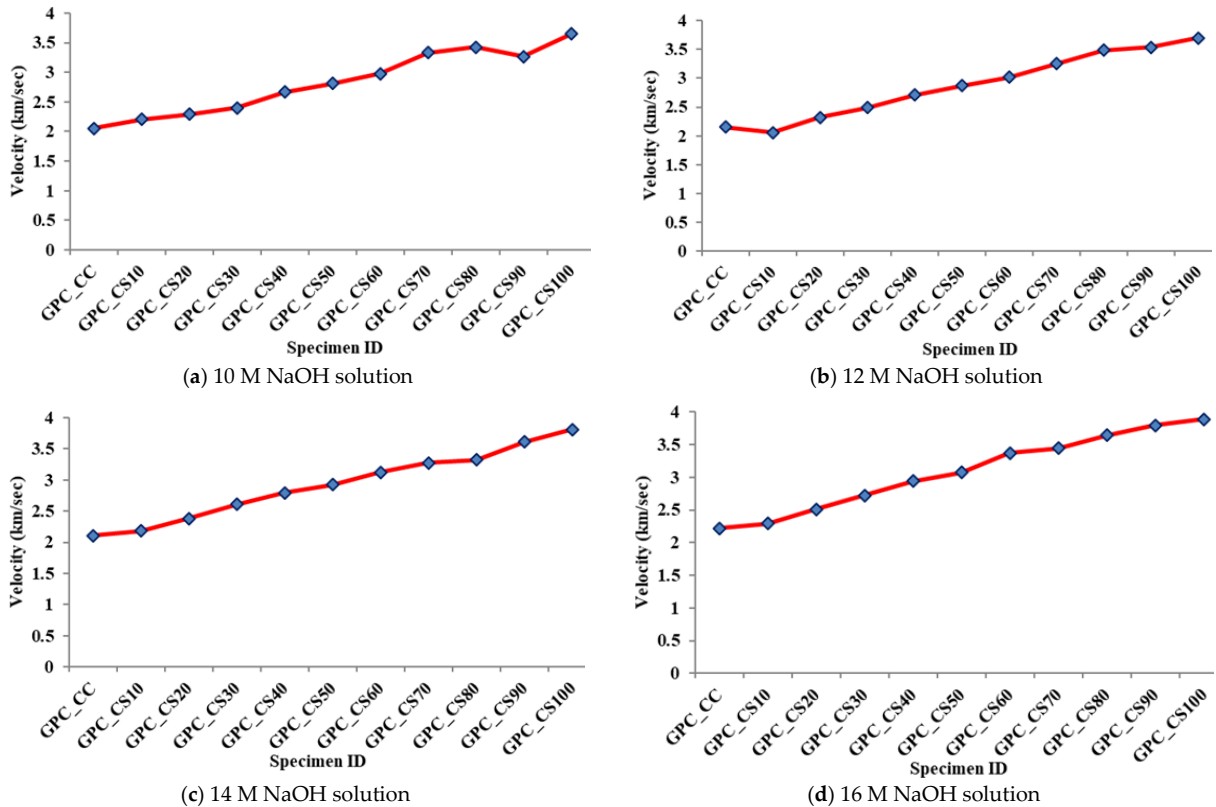

(**a**) 10 M NaOH solution        (**b**) 12 M NaOH solution

(**c**) 14 M NaOH solution        (**d**) 16 M NaOH solution

**Figure 10.** UPV results for GPC prepared with different NaOH molarities.

## 4. Microstructural Studies on GPC

*4.1. SEM Analysis*

The internal micro-mechanical characterization of ambient-cured GPC was performed using the scanning electron microscopy (SEM) analysis along with the elemental composition scan using Electron Dispersive Spectrum (EDS) analyses. The microstructure of the ambient-cured GPC with 100% copper slag and different NaOH molarities were captured at different resolutions and presented in Figure 11. From the figure, it can be seen that the copper slag GPC specimens contain numerous fly ash particles that are in the unreacted stage. Owing to the ambient curing process, a low value of latent heat at the initial curing timeline could have resulted in the delay of the polymerization reaction and formation of the Na-Al-Si-H gel [9]. The micrographs seem heterogeneous, compact and dense, indicating a high degree of cohesiveness between the binder phase and coarse particles, which explains the good mechanical performance [5,47]. Moreover, the polymerization reaction could have been delayed due to the water loss that occurred by the increase in the molarity of alkaline NaOH from 10 M to 16 M. From Figure 11a, a two-phased binder-aggregate interface of the developed GPC specimens can be seen. As specified, the GPC mix with copper slag contains three segments including (a) an aggregate, (b) a binder matrix and (c) an interfacial transition zone (ITZ). A strong pozzolanic-binder matrix is formed as a result of the reaction between the alkaline solution and copper slag which gets encapsulated over the surface of the aggregate [73,74]. At the magnification of 100 μm shown in Figure 11b, the occurrence of a dense ITZ can be observed which embodies the separation layer between the aggregate and polymeric matrix. The ITZ between the aggregate and geopolymer paste matrix was made stronger by the dissolution of the outer edges of the copper slag aggregate by the alkaline solution and holding them in place by hindering the polycondensation reaction with the matrix of the copper slag. These steps are responsible for forming a dense matrix, thereby improving the strength of the concrete [75,76].

From Figure 11c, it can be seen that the ambient-cured GPC had several smaller pores and cracks. Furthermore, the pores in the ambient-cured specimen were larger, as seen in the SEM images. These concrete pores could be the source of absorption. At later stages, it can be seen that the fly ash was in a well-reacted state and that the geopolymer paste matrix was visible. Moreover, the binder matrix was dense with a few partially reacted and unreacted fly ash particles present on it (Figure 11d). This unreacted fly ash presence may be due to the demand for the activator solution. In addition to the above-mentioned characteristics, the SEM images also contain a few cenosphere and plenosphere particles from the unreacted fly ash observed in the early stages of the ambient-cured copper slag GPC specimens.

*4.2. EDS Analysis*

Electron Dispersive Spectrum (EDS) analyses were carried out to determine the presence of major atomic constituents in the matrix following the reaction. Moreover, the EDS analysis was helpful in understanding the presence of various phases from the microstructure analysis. Figure 12 depicts the EDS results for the ambient curing of geopolymer concrete with various NaOH molarities. The test results indicate that the geopolymer concrete has major element traces of Al, Si, O and Na along with a few smaller peaks of Fe and Ca also noted in most of the EDAX spectra. The presence of predominant silica peaks is due to the existence of fly ash and its reaction with the alkaline activator. From Figure 12b,c, it can be seen that Fe peaks were also observed in the geopolymer concrete specimens created using the 12 M and 14 M NaOH solutions. Moreover, the main by-product due to the hydration reaction is the formation of Na-Al-Si-H gel which is responsible for the strength development of GPC.

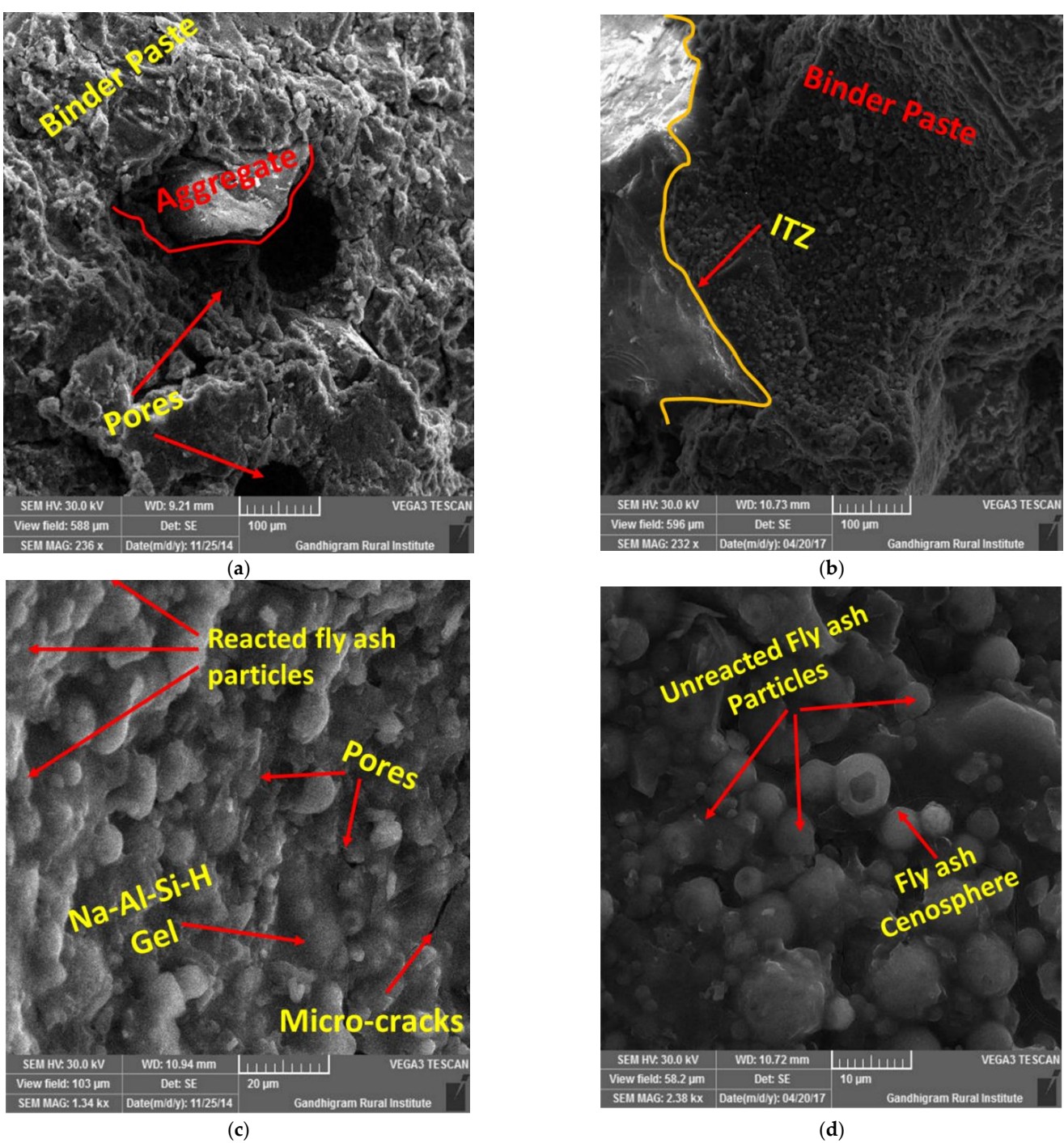

**Figure 11.** Microstructure analysis of GPC with copper slag under ambient curing.

### 4.3. FTIR Analysis

The Fourier Transfer Infrared (FTIR) spectroscopy was used to characterize the chemical bondage of the geopolymer materials reaction products. Figure 13 shows the FTIR spectrum of the ambient-cured GPC specimens with different NaOH molarity levels. Geopolymers are composed of Si-O tetrahedrons, which are connected via corner-sharing bridging

oxygen atoms. If the silica is in the amorphous state, the position of the main Si-O-X stretching band is approximately located at 1100 cm$^{-1}$. The asymmetric stretching vibrations at the 1050 cm$^{-1}$ to 1100 cm$^{-1}$ region show Si-O-Si (Al) bridges, which represent the degree of polymerization [52]. The bands in the 1550 cm$^{-1}$ to 1600 cm$^{-1}$ region correspond to the H-O-H bending vibration. Due to the molecular water, symmetric stretching vibrations of Si-O-Si were found at 778.22 cm$^{-1}$ in most of the specimens. The adsorption bands located in the 800 cm$^{-1}$–600 cm$^{-1}$ region represent the Al-O and tetrahedrally coordinated aluminum cation. From the FTIR spectral analysis, it was evident that the influence of copper slag did not affect the process of the polymeric reaction, but it enhanced the polymerization process. From the results, it was found that there was a stretching vibration of the O-H bond in the 3429 cm$^{-1}$ to 2300 cm$^{-1}$ region, a bending vibration of H-O-H in the 1595 cm$^{-1}$ to 1628 cm$^{-1}$ region and a stretching vibration of $CO_2$ located at about 1430 cm$^{-1}$. A peak was found at the 965 cm$^{-1}$ to 956 cm$^{-1}$ region, and these regions were assigned for the Si-O and Al-O of asymmetric stretching vibration related to the formation of geopolymer structure [77]. The band's position change is affected by the ash type, reactive glassy concentration of silica and alumina, curing temperature and, most importantly, curing time.

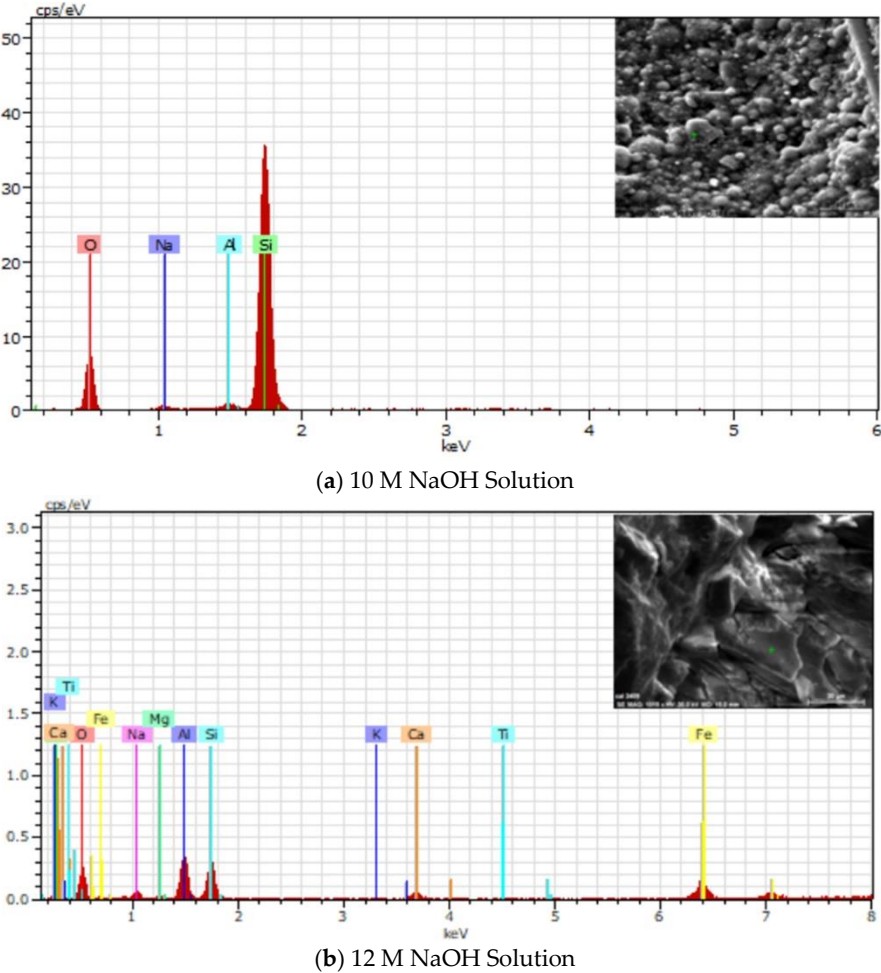

(**a**) 10 M NaOH Solution

(**b**) 12 M NaOH Solution

**Figure 12.** *Cont.*

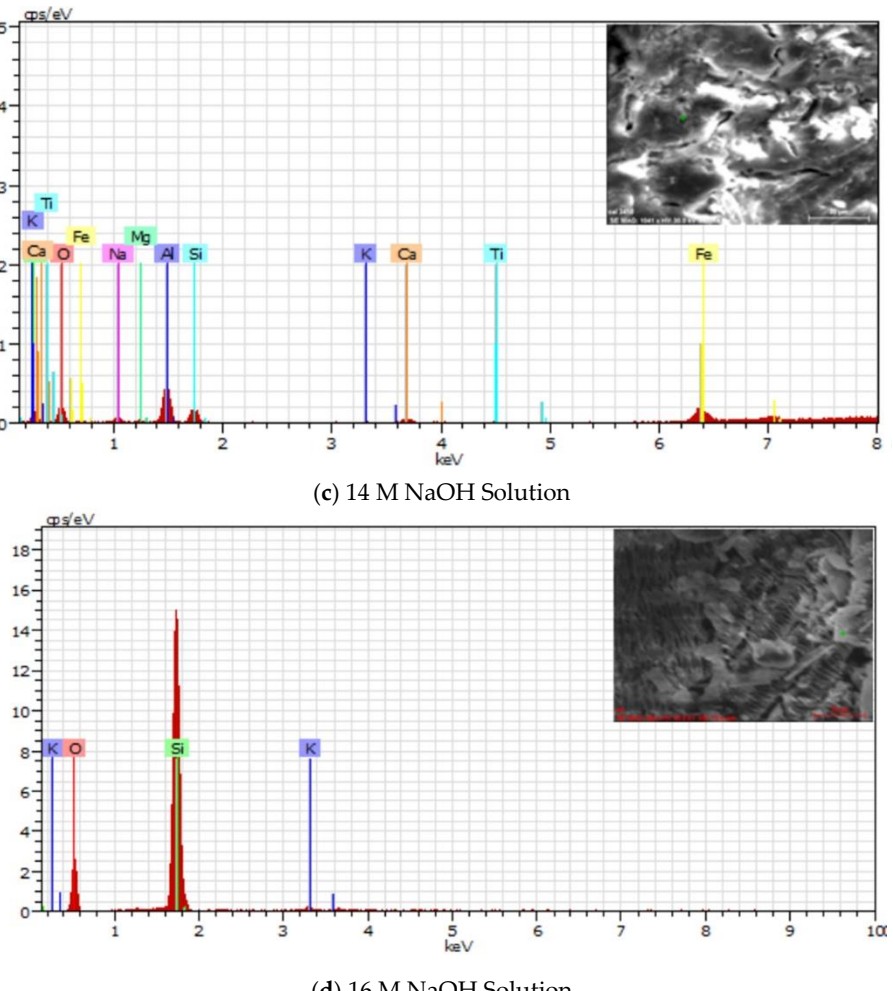

(**c**) 14 M NaOH Solution

(**d**) 16 M NaOH Solution

**Figure 12.** EDS analyses of GPC specimens with different NaOH molarities.

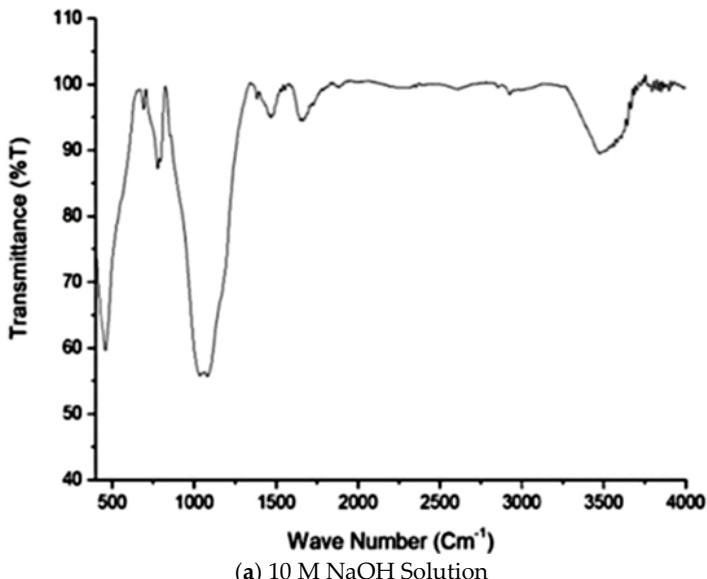

(**a**) 10 M NaOH Solution

**Figure 13.** *Cont*.

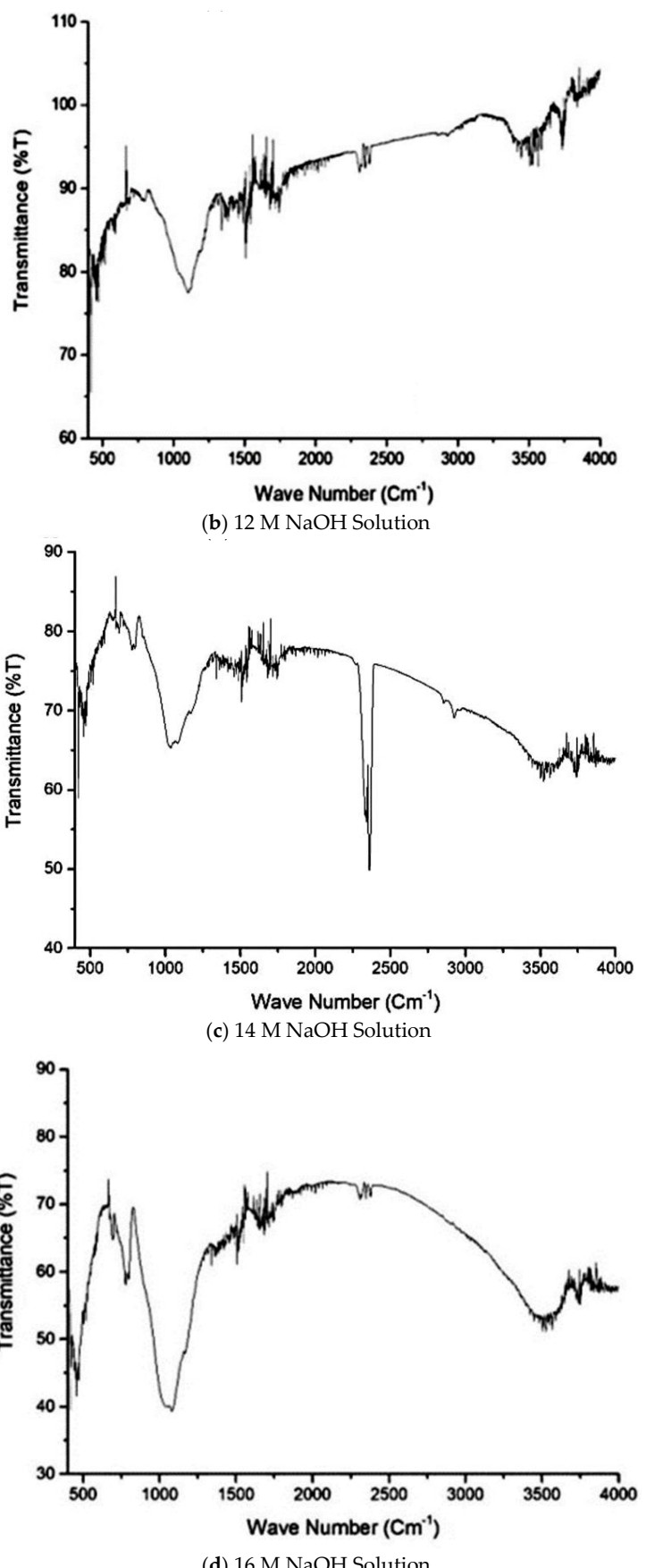

(**b**) 12 M NaOH Solution

(**c**) 14 M NaOH Solution

(**d**) 16 M NaOH Solution

**Figure 13.** FTIR Spectrum analysis for GPC specimens with different NaOH molarities.

## 5. Summary and Conclusions

In this study, the effect of developing sustainable copper slag-based geopolymer concrete with no conventional fine aggregate is explored. The purpose of the study was to understand the viability of using copper slag as a complete replacement for fine aggregate in the ambient curing of type PC without compromising the strength. In total, 44 mixes were evaluated for their strength and microstructural characterization. From the results obtained, the subsequent concluding remarks can be derived:

1. Replacement of river sand with 100% copper slag showed a gain in compressive, split tension and flexural strength by 53.8%, 66.9% and 39.3%, respectively, when compared to the control GPC with no copper slag (CC10). Similar findings were observed for different molarity levels of NaOH (10 M, 12 M, 14 M and 16 M) with no detrimental effects on the mechanical properties.
2. The use of different molarities of the NaOH solution yielded positive results on the tensile and compressive strength enhancement of ambient-cured geopolymer concrete. For the control specimen with 16 M NaOH molarity, the increase in compressive, split tensile and flexural strength was found to be 11.2%, 28.2% and 13.2%, respectively, when compared to the CC10 specimen. Similarly, for the GPC specimen with the 16 M NaOH solution and 100% copper slag, the increase in compressive, split tensile and flexural strength was found to be 31.9%, 41.7% and 11.8%, respectively, when compared to the G10CS100.
3. Microstructural characterization of the GPC specimens using SEM analyses revealed the presence of unreacted fly ash particles and pores at the initial stages of curing. However, most fly ash particles were found to have reacted after 28 days of ambient curing. Results from the EDS analyses showed that the element traces of Si, Al, O and Na had the major predominant peaks observed in the spectrum due to the formation of Na-Al-Si-H gel. In addition to the above-mentioned major elements, smaller peaks of Fe and Ca were also noted from the EDS analyses of the GPC samples with different NaOH molarity levels.
4. The analytical predictions from the Euro code (EC-04) showed a close correlation with the experiments for split tensile strength and flexural strength. However, they overpredicted the values of the elastic modulus. The experimental values of the elastic modulus for GPC were closely captured using the equation suggested by the CSA-04.
5. The preliminary findings of this work provide an understanding of the mechanical and microstructure characterization of sustainable GPC produced by replacing 100% river sand with copper slag. The results will be of significant value as the usage of copper slag at higher volumetric replacement seems to have no adverse effects on the strength characteristics of GPC, which supports its use as an alternate material to promote sustainable construction technology. However, it is highly essential to examine the durability characteristics and to account for the long-term performance of high-volume copper slag-based GPC. These additional durability studies are recommended as the scope for further work.

**Author Contributions:** Conceptualization, N.A. and M.C.; Methodology, N.A., M.C. and T.O.; Investigation, N.A. and J.M.; Validation, N.A. and J.M.; Formal analysis, N.A.; Data curation, J.M. and T.O.; Resources, J.M.; Writing—original draft preparation, N.A.; Writing—review and editing, M.C., J.M. and T.O.; Visualization, M.C. Funding acquisition, M.C. All authors have read and agreed to the published version of the manuscript.

**Funding:** This research received no external funding.

**Institutional Review Board Statement:** Not applicable.

**Informed Consent Statement:** Not applicable.

**Data Availability Statement:** The data related to the present study are available in the manuscript.

**Conflicts of Interest:** The authors declare no conflict of interest.

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
