# Peer review of "Effective Utilization of Copper Slag for the Production of Geopolymer Concrete with Different NaOH Molarity under Ambient Curing Conditions"

_sustainability, doi:10.3390/su142316300_

Round 1

Reviewer 1 Report

This paper explores the effect of developing sustainable copper slag based geopolymer concrete with no conventional fine aggregate. The experiment explores the fresh (slump) and hardened concrete properties by controlling the NaOH molarity (10, 12, 14 and 16) and the replacement levels of copper slag ranging from 0 to 100%. The viability of using copper slag as a complete replacement for fine aggregate in ambient curing type PC without compromising the strength is well illustrated by evaluating the strength and micro-structural characterization of 44 mixes. It was concluded that a cohesive and fully compact geopolymer matrix was achieved together with the use of low-calcium fly ash and copper slag under ambient curing conditions. In this paper, the research idea is clear and the content is rich. The description of the experimental process is detailed and complete, and the method used is scientific and reasonable. Through the detailed analysis of the microstructure of mixture, the conclusion is more convincing. However, the content of the paper needs great revision to improve the quality. The narrative content is too cumbersome and structure is not compact enough. The language is not concise and accurate enough. The discussion is so long or complicated that it is difficult to interest readers. The introduction is confusing, lengthy and cumbersome. In addition, the following questions also need the author’s attention:

(1) Please note to modify the article format, with two blank spaces at the beginning of each paragraph to facilitate readingï¼› (2) The introduction part describes too much about the research background of the paper and the author can delete it appropriately. The author should highlight the significance and scientific value of this studyï¼› (3) Underscore the scientific value-added to your paper in your abstract. What problem are we looking at? why are we solving this problem? That would help a prospective reader of the abstract to decide if they wish to read the entire paperï¼› (4) The second and third chapters of the article can be merged with the first chapter to streamline the languageï¼› (5) The experiment and analysis part of the paper is too cumbersome and confusing. Can it be simplified and one-to-one correspondence? (6) The conclusion part is an important part of the paper, which is a summary of the research work. Highlight the novelty of your study. Clearly discuss what the previous studies. What are the Research Gaps/Contributions ?

Author Response

Dear Reviewer,

We are very grateful for the insightful comments provided to improve the quality of this manuscript. The reviews were encouraging and have been considered in the revised manuscript. The attached file contains the detailed response for the comments raised.

Thank you.

Reviewer 2 Report

Kindly refer and address the comments given in the attached file.

Author Response

(The authors gave the same response as above.)

Reviewer 3 Report

An extensive and analytical paper on geopolymer concrete. 

Authors use different molarities of alkaline solutions for the same mix proportions and they use the same specimen ID for specimen produced with different molarities. This should be avoided for tracibility reasons.

Be careful with the abbreviations. I am not sure if SP in mix proportions stands for superplasticizer since it is not mentioned before.

There are two "table 2"

There are no statistics for the test. Standard deviation is not mentioned and there are no bars on charts. This should be taken in mind, since for example in figure 6 authors attempt to compare experimental values with theoritical ones.

Author Response

(The authors gave the same response as above.)

Reviewer 4 Report

This work evaluated the effect of the replacement of the copper slag and different NaOH molarity under ambient curing conditions. The results displayed in this work were very systematic and comprehensive. However, the explanation of the geopolymerization mechanism was relatively thin, which could give more support to disclose the interactive relationship between the properties of resulting GPC with the introduction of the copper slag and different NaOH molarity.

Author Response

Dear Reviewer,

Thank you very much for providing a deep and thorough review. We are very grateful for the insightful comments provided to improve the quality of this manuscript. The reviews were encouraging and have been considered in the revised manuscript. The attached file contains the detailed response for the comments raised.

Round 2

Reviewer 1 Report

This study investigates the physical and micro-structure characterization of sustainable geopolymer concrete (GPC) developed with copper slag as a replacement for fine aggregate. In total, forty-four geopolymer concrete mixtures were prepared to examine their fresh and hardened properties. Four different NaOH molarity (10, 12, 14 and 16) and the replacement levels of copper slag ranging from 0 to 100% with an increase of 10% were considered as a variable in this research. The study parameters examined include the fresh (slump) and hardened concrete properties. The viability of using copper slag as a complete replacement for fine aggregate in ambient curing type PC without compromising the strength is well illustrated by evaluating the strength and micro-structural characterization of 44 mixes.It was concluded that a cohesive and fully compact geopolymer matrix was achieved together with the use of lowcalcium fly ash and copper slag under ambient curing conditions.

Presented study is interesting and may constitute a valuable contribution to the area of knowledge.

At the same time, I would like to note that the manuscript has a number of minor shortcomings, the elimination of which will improve the scientific quality and increase the perception of presented information.

(1) From my point of view, the literature review is rather poorly presented in the text of previous studies.

1.1) The introduction is too lengthy and confusing.The author can use refined sentences to describe what other scientists have previously revealed, and should focus on the scientific value and research significance of this study.

1.2) It is necessary to more clearly outline the problems raised by the authors. The second chapters of the article can be merged with the first chapter to streamline the language.

1.3) In lines 67-68; 71-73; 456-458 the authors make a number of statements that are not confirmed by the corresponding references.

(2) In lines 67-68, the author mentions the problem of collecting sand as concrete at the beach, which is controversial.

(3) There are obvious format errors in lines 141, 147, 265.

(4) Figure 1 (a) is of poor quality, it is recommended to rebuild it.

(5) English sentences are not smooth,it is suggested to amend them appropriately.

(6) In the discussion of test results, the author only described the test results, but did not explain the reasons for the results. It is suggested to add corresponding analysis to better reflect the scientific significance of this study.

(7) It is necessary to reflect the results obtained on the basis of the analytical work carried out, your own conclusions on the problem under consideration. It is necessary to work on the conclusions and present more specific ones, obtained as a result of analysis and deductions.

Author Response

Dear Reviewer,

Once again, thank you very much for providing a deep and thorough review. We are very grateful for the insightful comments provided to improve the quality of this manuscript. The reviews were encouraging and have been considered in the revised manuscript. The attached file contains the detailed response for the comments raised.

Reviewer 2 Report

All the comments were answered by the authors 

Author Response

Dear Reviewer,

Once again, thank you very much for providing a deep and thorough review. We are very grateful for the insightful comments provided to improve the quality of this manuscript. The reviews were encouraging and have been considered in the revised manuscript.
